# The Association of Malnutrition and Health-Related Factors among 474,467 Older Community-Dwellers: A Population-Based Data Mining Study in Guangzhou, China

**DOI:** 10.3390/nu16091338

**Published:** 2024-04-29

**Authors:** Wei-Quan Lin, Ting Xiao, Ying-Ying Fang, Min-Ying Sun, Yun-Ou Yang, Jia-Min Chen, Chun-Quan Ou, Hui Liu

**Affiliations:** 1Department of Basic Public Health, Guangzhou Center for Disease Control and Prevention, Guangzhou 510440, China; linweiquan0503@163.com (W.-Q.L.); gzcdc_fangyy@gz.gov.cn (Y.-Y.F.); sunmy1220@163.com (M.-Y.S.); gzcdc_chenjm@gz.gov.cn (J.-M.C.); 2Institute of Public Health, Guangzhou Medical University & Guangzhou Center for Disease Control and Prevention, Guangzhou 510440, China; 3State Key Laboratory of Organ Failure Research, School of Public Health, Southern Medical University, Guangzhou 511436, China; xiaoting9720@163.com

**Keywords:** prevalence, health-related factors, patterns of malnutrition, data mining, older community-dwellers

## Abstract

Background: This study aimed to examine the prevalence and associated factors of malnutrition in older community-dwellers and explore the interaction between associated factors. Methods: A total of 474,467 older community-dwellers aged 65 or above were selected in Guangzhou, China. We used a two-step methodology to detect the associated factors of malnutrition and constructed logistic regression models to explore the influencing factors and interactive effects on three patterns of malnutrition. Results: The prevalence of malnutrition was 22.28%. Older adults with both hypertension and diabetes (RERI = 0.13), both meat or fish diet and hypertension (RERI = 0.79), and both meat or fish diet and diabetes (RERI = 0.81) had positive additive interaction effects on the risk of obesity, whereas those on a vegetarian diet with hypertension (RERI = −0.25) or diabetes (RERI = −0.19) had negative additive interaction effects. Moreover, the interactions of physical activity with a meat or fish diet (RERI = −0.84) or dyslipidemia (RERI = −0.09) could lower the risk of obesity. Conclusions: Malnutrition was influenced by different health factors, and there were interactions between these influencing factors. Pertinent dietary instruction should be given according to different nutritional status indexes and the prevalence of metabolic diseases to avoid the occurrences of malnutrition among older adults.

## 1. Introduction

The aging population is growing at an unprecedented pace worldwide. The proportion of people aged 65 years or above globally is projected to increase from 9.3% to 16.0% between 2020 and 2050 [1]. China has and will have the largest older population in the world [2]. In 2020, there were 190.64 million people aged 65 or above in China, approaching nearly 14.0% of the total population [3]. With the continuous increase in the older population, problems of health and life quality related to older adults have become important social issues. Preventing malnutrition among older adults has been identified as one of the best strategies for achieving healthy aging [4].

Malnutrition could lead to a series of health issues in the older population, such as a decline in general functional status, impaired immunity, prolonged hospitalization, a rise in healthcare costs, and increased risk of morbidity and mortality, which not only influence the well-being and life quality of the individual but also impose heavy economic burdens on families and society [5,6,7]. On the contrary, adequate nutrition could prevent, delay, or significantly improve a large number of chronic diseases affecting older adults, which indicates that nutritional interventions for older adults could be useful in promoting healthier and more active aging [8]. As a result, it is necessary to identify the prevalence and determinants of malnutrition and work to improve the nutrition status of older adults who are at risk of malnutrition.

There is no consistent definition for malnutrition at present [9]. Most studies focus on deficiencies in energy and/or nutrient intake, such as undernutrition and inadequate vitamins or minerals, and the terms undernutrition and malnutrition were often used interchangeably [10]. However, according to the definition of malnutrition from WHO [11], malnutrition also includes excesses or imbalances in a person’s intake of energy and/or nutrients, such as overweight and obesity. The double burden of malnutrition and diet-related non-communicable diseases brought challenges to a majority of low- and middle-income countries [12]. Therefore, this study considered a more comprehensive form of malnutrition, which encompassed both overnutrition (obesity) and undernutrition (underweight and low hemoglobin) [13]. The prevalence of malnutrition among the Chinese older adults in the community reported in previous studies was at a high level and varies considerably, ranging from 12.6% to 52.9% [14,15]. One of the reasons for the variation is that the prevalence of malnutrition changes over time [16], which means that an updated report needs to be issued. Additionally, other differences in factors such as assessment criteria, sample size, and study region may also result in variation [17]. The high prevalence of malnutrition and the subsequent serious health complications damage the quality of life and cause great losses to the healthcare system [8], indicating the urgency to explore malnutrition, particularly the potential risk factors.

The aging process was not the only factor determining malnutrition. Many previous studies explored the effects of other factors, such as sociodemographic factors, chronic diseases, and psychosocial factors on malnutrition in older adults [9]. However, these studies did not consider the interactions between health-related factors that were likely to improve the risk of malnutrition in older adults. Clarifying these factors would help with specific control measures, since many factors are modifiable.

Therefore, based on a large representative sample, we aimed to assess the prevalence of malnutrition and its association with health-related factors for the older community-dwelling population. Additionally, the effects of interactions between health-related factors on three patterns of malnutrition were explored. This study might supplement evidence for standardizing the intervention and management of malnutrition, which could contribute to improving the nutritional status of older adults.

## 2. Materials and Methods

### 2.1. Study Design and Participants

The population of this study was the older adults who participated in the health management service project at 186 community healthcare centers in Guangzhou from January to December 2020.

Health management service for older adults is one of the national basic public health services projects that are provided free of charge by the Chinese government. It is implemented by community healthcare centers where older community dwellers aged ≥ 65 years are entitled to have a free physical examination service every year. By 2020, Guangzhou has established 186 community healthcare centers in all streets and towns to provide primary healthcare services for older adults (Appendix A).

### 2.2. The Monitoring System of Health Management Service for Older Adults

The monitoring system is constructed based on individual electronic health records of older adults from all community healthcare centers, including socio-demographic characteristics and health-related factors.

#### 2.2.1. Socio-Demographic Characteristics

The socio-demographic characteristics, including age, gender, census register, living areas, education level, marital status, and medical insurance, were selected for the study.

#### 2.2.2. Health-Related Factors

Health-related factors are considered lifestyle behaviors and underlying diseases. The lifestyle behaviors included current smoking, alcohol consumption, physical activity, and dietary habits. Current smoking or alcohol consumption was defined as whether someone smoked or drank in the past year. Physical activity was measured as whether someone performed exercise regularly. Dietary habits were divided into three categories: balanced diet, meat or fish diet, and vegetarian diet using a self-assessment item about the consumption of meat, fish, and vegetables. Specifically, a balanced diet was defined as a daily intake of meat, fish, and vegetables, mostly in accordance with the requirements of the food pyramid [18]; a fish or meat diet was an average of more than 50 g of meat and fish consumed per day [19]; a vegetarian diet included consuming eggs, milk, and dairy products but not consuming any meat or fish, or eat at most once a month [20]. In addition, hypertension, diabetes, and dyslipidemia were three common chronic diseases among older adults [21], which not only affected the health status of older adults but had also been shown to be associated with their nutritional status in previous studies [22,23,24]. The underlying diseases, including hypertension, diabetes, and dyslipidemia, which were detected by self-report or physical examination, were included in this study to explore the impact of these diseases on the nutritional status of older adults in order to enhance their overall health.

#### 2.2.3. Malnutrition Measurement

Based on Body Mass Index (BMI) and hemoglobin (Hb), we considered three patterns of malnutrition: underweight, obesity, and low hemoglobin [11]. According to the Chinese BMI reference standards, participants were categorized as underweight (<18.5), normal weight (18.5–23.9), overweight (24.0–27.9), and obese (≥28.0). Additionally, an Hb of males (adults) less than 120 g/L or that of females (adults) less than 110 g/L was defined as low hemoglobin.

### 2.3. Ethics Approval

Ethical approval for this survey was obtained from the Ethics Committee of the Guangzhou Center for Disease Control and Prevention (GZCDC-ECHR-2020P0004).

## 3. Statistical Analysis

The socio-demographic characteristics and health-related factors of the older adults were summarized as counts and percentages. Prevalence estimates of malnutrition and its three patterns (underweight, obesity, and low hemoglobin) were calculated separately for the overall population and subgroups stratified by age and gender. We did not impute missing data due to the highest missing rate of the factors was 3.7%.

A two-step analysis strategy was proposed to explore the correlations between malnutrition patterns and health-related factors among older community-dwelling people. First, association rule mining of the A priori algorithm was used to generate the hypothesis and extract association rules of the form *H* → *M*, where *M*s were three patterns of malnutrition, and *H*s were health-related factors and their interactions. The associations were evaluated by the measures of Support (%), Confidence (%), and Lift. Additionally, the final association set was filtered and only included the rules that satisfied the criteria of Support > 2%, Confidence > 5%, and Lift > 1.0. Second, the multivariate logistic regression analysis was used to further validate and quantify the association rule using odd ratios (*OR*s) and their 95% confidence intervals (CIs). Among them, multinomial logistic regression was performed for BMI (underweight and obesity) and binomial logistic regression was performed for low hemoglobin. Three logistic regression models, including all health-related factors in the final association set, were constructed for each outcome (three patterns of malnutrition). The first model was unadjusted for any confounding factors, the second was adjusted for age and gender, and the third was adjusted for all socio-demographic characteristics, including age, gender, census register, living areas, education level, marital status, and medical insurance. Multiplicative interaction effects were detected by OR while additive interaction effects were evaluated using three measures [25]: the relative excess risk due to interaction (RERI), the attributable proportion due to interaction (AP), and the synergy index (S). If there exists a categorical variable for which the majority of individuals choose the same response value, it may lead to the perfect separation problem. To address this type of variable, a model that included and excluded this variable was separately constructed and compared. The variable would be retained in the model if the results did not differ significantly. A *p* < 0.05 was considered to be statistically significant in each analysis and all the statistical analyses were performed using R, version 4.1.1.

## 4. Results

### 4.1. Participant Selection

Figure 1 illustrates the flow chart of participant selection in the study. From January to December 2020, 515,106 older adults participated in the health management service project, of whom 40,639 were excluded because of duplicate or missing outcome data. Finally, one-third (474,467/1,460,333) of the older population in Guangzhou was included in the present study, which was indicated as a large representative sample.

### 4.2. Participant Characteristics

Table 1 shows the distribution of socio-demographic characteristics and health-related factors for all the participants. In the study, 105,717 (22.28%) were classified as malnutrition, of whom 21,379 (4.51%) were underweight, 52,665 (11.10%) were obese, and 38,093 (8.03%) had low hemoglobin. We divided the participants into five groups according to age, and the ages of participants were predominantly concentrated in the 65–69 years age range (40.33%). Additionally, more female participants were included in the study compared to male participants (58.79% vs. 41.21%). The distribution of age and gender of the study population were similar to those of the seventh national population census of Guangzhou (Appendix A).

### 4.3. Prevalence of Malnutrition and Its Three Patterns

Table 2 displays the prevalence of malnutrition and its three patterns. The overall prevalence of malnutrition, underweight, obesity, and low hemoglobin were estimated to be 22.28% (95% CI: 22.16–22.40), 4.51% (95% CI: 4.45–4.57), 11.10% (95% CI: 11.01–11.19), and 8.03% (95% CI: 7.95–8.11), respectively. Among them, the prevalence of malnutrition increased with age, and males had a higher prevalence of malnutrition (24.05% vs. 21.04%), underweight (4.58% vs. 4.45%), and low hemoglobin (11.15% vs. 5.84%) while they had a lower prevalence of obesity (10.14% vs. 11.77%) compared with females.

### 4.4. Potential Influencing Factors and Interactions

Table 3 lists the potential influencing factors and interactions detected by association rule mining. After pruning and filtering based on the minimum criteria, there remained two association rules for underweight, twelve for obesity, and three for low hemoglobin. Physical activity and its interaction with dietary habits might influence the probability of being underweight. In addition, obesity was strongly correlated with diabetes, hypertension, dyslipidemia, physical activity, and the interactions of these four factors with dietary habits. The effect of the interactions of hypertension with diabetes, dyslipidemia, or physical activity on obesity was also observed. Obesity has also been linked to the interaction of dyslipidemia with physical activity. Physical activity, diabetes, and dietary habits were detected to be associated with low hemoglobin, while no interaction was found.

### 4.5. Association Factors Validated by Logistic Regression

Figure 2 shows the logistic regression results for three patterns of malnutrition. Whether the model controlled for confounders had little effect on the relationship between three patterns of malnutrition and health-related factors. Physical activity showed a protective effect on developing underweight (OR = 0.86, 95% CI: 0.84–0.89) and low hemoglobin (OR = 0.64, 95% CI: 0.62–0.65), whereas the older adults who performed physical activity were more prone to be obese (OR = 1.04, 95% CI: 1.02–1.06). A Meat or fish diet was a risk factor for obesity (OR = 1.92, 95% CI: 1.72–2.15), but on the other hand, it was a protective factor for low hemoglobin (OR = 0.72, 95% CI:0.61–0.84). Additionally, the older adults with hypertension, diabetes, and dyslipidemia were less likely to be underweight (hypertension: OR = 0.81, 95% CI: 0.78–0.83; diabetes: OR = 0.85, 95% CI: 0.82–0.89; dyslipidemia: OR = 0.81, 95% CI: 0.78–0.83) but more likely to be obese (hypertension: OR = 1.39, 95% CI: 1.36–1.43; diabetes: OR = 1.23, 95% CI: 1.21–1.26; dyslipidemia: OR = 1.18, 95% CI: 1.15–1.20). Furthermore, patients with diabetes were prone to suffer from low hemoglobin (OR = 1.12, 95% CI: 1.10–1.15).

Table 4 displays the interaction effects between health-related factors for underweight and obesity. Two chronic diseases had positive additive interaction effects on underweight older adults (hypertension and dyslipidemia: RERI = 0.18, AP = 0.30, S = 0.67; hypertension and diabetes: RERI = 0.11, AP = 0.15, S = 0.73; diabetes and dyslipidemia: RERI = 0.13, AP = 0.18, S = 0.72). Older people with both hypertension and diabetes were at high risk for obesity (RERI = 0.13, AP = 0.08, S = 1.23). Moreover, a meat or fish diet with hypertension (meat or fish diet and hypertension: RERI = 0.79, AP = 0.27, S = 1.72) or diabetes (meat or fish diet and diabetes: RERI = 0.81, AP = 0.28, S = 1.74) in these older people also increased the risk of obesity, but a vegetarian diet could reduce the probability (vegetarian diet and hypertension: RERI = −0.25, AP = −0.18, S = 0.61; vegetarian diet and diabetes: RERI = −0.19, AP = −0.16, S = 0.48). Moreover, the interactions of physical activity with other factors, including meat or fish diet (RERI = −0.84, AP = −0.53, S = 0.42) and dyslipidemia (RERI = −0.09, AP = −0.07, S = 0.73), could lower the risk of developing obesity.

### 4.6. Sensitive Analysis

A total of 97.49% of the individuals chose balanced dietary habits. Logistic regression models were constructed for the inclusion and exclusion of dietary habits, and the effects of other influencing factors were consistent with both models (Appendix A). The results of the stepwise regression are shown in Appendix A.

## 5. Discussion

This was the first large-scale cross-sectional study to investigate health-related factors and their interactions associated with different patterns of malnutrition risk among community-dwelling older adults in China. Older adults who lack medical resources might be more willing to participate in free physical examinations in primary health care. This study revealed their malnutrition status and explored related influencing factors, providing strategies to help improve health status and reduce medical expenses. The age and gender structure of participants in the research were similar to the Guangzhou population, which could represent the Guangzhou older population, and was comparable with the results of other community-based research. In the current study, 22.28% of the community-dwelling older adults suffered from malnutrition. A meta-analysis included 13 studies on the prevalence of malnutrition in China over the past decade showing that the combined prevalence of malnutrition in the community-dwelling older population was 14.1% [16], and another study reported that the global pooled prevalence of malnutrition in the older community varied from 0.8% to 24.6% [26]. The differences between these findings might be caused by differences in race, economic development levels, survey time, and assessment criteria, but still indicate the high level of malnutrition in the community-based older population in Guangzhou.

The prevalence of malnutrition increased with age, which was consistent with previous investigations [17,27,28]. Researchers have revealed that as the body aged, there were changes in body composition, declining gastrointestinal function, and reduced feeding drive, which might also show an impact on digestion and absorption, leading to an increased risk of malnutrition [17,29]. In addition, aging is often accompanied by the loss of lean muscle mass and increased risk of osteoporosis, which might limit the activities of older adults and make it more difficult to prepare food and even eat [30]. Males were more prone to suffer from malnutrition than females in this sample, and a previous study among older adults in Taiwan, China had the same results [4]. Whereas there was also evidence suggesting that the prevalence of malnutrition was higher among females [16,31,32], factors such as the economic independence of females and their status in family and society might have impacts on their nutritional status. There were still controversies over the way how gender affected malnutrition, and further studies were needed to explore and explain the issue.

Separately, we compared the results of the present research with other research in China with the same evaluation criteria and research subjects (community-dwelling older adults over 65 years). The incidence of underweight in this study was similar to the local findings reported from Shanghai, Eastern China (4.8%) [33] and Hubei Province, Central China (4.8%) [34] but higher than Shenzhen, Southern China (3.69%) [35]. And the incidence of obesity in the current findings was between the local findings reported from Shanghai (13.8%), Hubei (10.1%), and Shenzhen (9.27%). Blood hemoglobin has been found to be a useful marker of adult malnutrition [36]. A higher incidence of low hemoglobin was reported among older adults in the current study compared to previous studies reported from Chengdu, Southwest China (3.1%) [37], Haining, Southeast China (6.4%) [38], and Yangzhou, Eastern China (7.4%) [39] but a lower incidence reported from Wuhan, Central China (17.91%) [40]. The differences in prevalence rates among these studies might have been caused by differences in the economic level of the city and the demographic characteristics of the study participants. In addition, researchers have found that the incidence of anemia and underweight showed a downward trend over time [22]. Thus, the differences in survey implementation time explained the differences in prevalence rates.

The current study found that physical activity had a protective effect on developing underweight and low hemoglobin, which showed that exercise could improve nutritional status. Due to the metabolic changes resulting from aging, undernutrition was often followed by the preferential loss of fat-free mass in older adults, while physical exercise exerted a positive influence on anabolism and might promote the effect of renutrition in malnourished older patients [41]. On the one hand, physical activity could have a positive anabolic effect on muscle protein; on the other hand, physical activity might facilitate appetite and food intake among older adults at risk of undernutrition [42]. Moreover, the meat or fish diet was a risk factor for obesity and it was also observed in cohort studies that meat-eaters were more likely to be obese than other diet groups [43,44]. However, the present study showed that physical activity increased the risk of obesity, which was against the viewpoint supported by strong evidence that regular exercise contributed to body weight and fat loss [45]. One possible explanation was that fat people performed more exercise to lose weight rather than physical activity increased weight gain. The Health Belief Model (HBM) assumed that individuals perceived the seriousness and potential consequences of the condition and took specific health behaviors, which meant that fat persons perceived the threat of the risk of obesity might, in ways of diet control or physical activity, manage their weight [46]. In addition, the interactions of physical activity with a meat or fish diet could lower the risk of developing obesity in the present study, which could also support the above point of view. The key points of nutrition in older adults were a healthy diet and appropriate physical exercise. Notably, physical activity could alleviate the effects of a poor diet on the nutritional status of old people.

Another finding of the present study was that diabetes, hypertension, and dyslipidemia were positively associated with obesity. Along with obesity, both diabetes and cardiovascular diseases were associated with overnutrition [22]. Diabetes, hypertension, and dyslipidemia were common complications of obesity. This could be partly explained by the fact that diabetic patients had disorders in the metabolism of the three major nutrients due to internal dysfunction of their glands [47,48]. Moreover, it was hard for diabetic patients to conduct dietary management and have proper nutrient ratios [49]. Our participants with hypertension were more likely to be obese. The underlying reason might be the pathophysiological effects of diseases, which lead to metabolic abnormalities [50]. Evidence indicated that abnormal blood lipids might lead to insulin resistance in peripheral tissues, which in turn affects fat metabolism in the liver after eating, leading to overweight and obesity [23]. Additionally, we found that older people with diabetes were more likely to show symptoms of low hemoglobin than non-diabetics, which was consistent with previous studies [24,35]. Research indicated that the incidence and prevalence of diabetic patients were often related to erythropoietin deficiency caused by diabetic kidney damage [51]. The interaction of heart failure, renal dysfunction, anemia, and iron deficiency form a vicious cycle and diabetes further accelerates this vicious cycle [52]. Therefore, diabetic patients might be more likely to suffer from low hemoglobin symptoms.

We explored the interaction between underweight and obesity with health-related factors and found that older people with both hypertension and diabetes were at a higher risk of obesity. Other researchers have also reported that people with two or more chronic diseases were more likely to suffer from malnutrition [53]. Considering the high burden of chronic disease and comorbidity, it was crucial to aggressively control weight within a reasonable range and improve the quality of life in older patients. In addition, a meat or fish diet could further increase the risk of obesity among older adults with hypertension or diabetes, whereas a vegetarian diet could reduce this risk. The nutritional status of older adults is the result of complex interactions between dietary, socioeconomic, physical, and psychological factors [54]. A healthy diet helps prevent or control many important diseases, especially obesity, malnutrition, and diabetes [55]. A vegetarian diet was an independent protective factor of low hemoglobin, whereas it did not show an independent effect on obesity. However, the interaction effect of a vegetarian diet and hypertension or diabetes had a protective effect on obesity in older adults. The result suggested that pertinent dietary instruction should be given according to different nutritional status indexes and the prevalence of metabolic diseases to adjust the intake of nutritional components so as to avoid the occurrences of malnutrition among old people.

## 6. Strengths and Limitations

This study was based on a large representative sample covering one-third of the older population in Guangzhou and revealed their nutritional status. The impact of health-related factors and their interactions on three patterns of malnutrition were investigated by data mining, which might supplement evidence for standardizing the intervention and management of malnutrition. Despite its strengths, there were several limitations in the present study. First, the cross-sectional design of this study only can infer correlations but cannot show the causal relationships between the response and explanatory variables. Second, the study was conducted in a city with a good economic development level, which might not represent all the regions of China. However, the age structure of the participants was close to “the Seventh National Population Census in 2020” and the results can provide a reference for the nutritional level of Chinese older adults. Third, some important potential influencing factors of malnutrition were not collected in the survey, which might cause bias or misleading results; more comprehensive consideration will be given in future research.

## 7. Conclusions

The level of malnutrition was high and the problem of overnutrition and undernutrition both existed among the community-based older population in Guangzhou, which were worth paying more attention to. The prevalence of malnutrition increased with age and males had a higher prevalence of malnutrition compared to females. Meat or fish diet and dyslipidemia were positively associated with overnutrition, while physical activity and vegetarian diet showed positive effects on undernutrition. Furthermore, the interactions of physical activity with other factors, including meat or fish diet and dyslipidemia, could lower the risk of developing obesity. Older people with hypertension or diabetes were at high risk of obesity and a meat or fish diet in these older adults might further increase the risk, whereas a vegetarian diet could reduce the hazard. In a word, pertinent dietary instruction should be given according to different nutritional status indexes and the prevalence of metabolic diseases to adjust the intake of nutritional components so as to avoid the occurrences of malnutrition among old people. These findings might help in developing and implementing targeted interventions based on different types of malnutrition.

## Figures and Tables

**Figure 1 nutrients-16-01338-f001:**
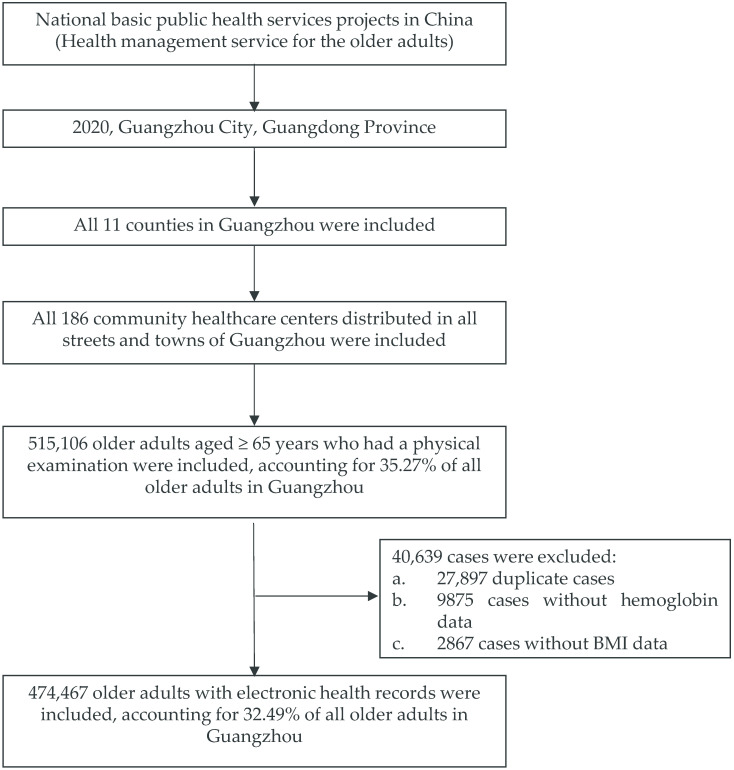
Data framework of the study in 2020 in Guangzhou, China.

**Figure 2 nutrients-16-01338-f002:**
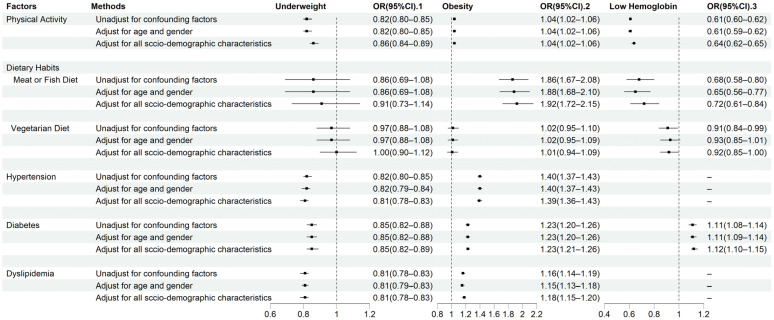
Association factors with three patterns of malnutrition explored by multivariate logistic regression (Main effect model).

**Table 1 nutrients-16-01338-t001:** Socio-demographic characteristics and health-related factors for all the participants.

Characteristics	Overall *n* (%)	Malnutrition *n* (%)	Underweight *n* (%)	Obesity *n* (%)	Low Hemoglobin *n* (%)
	*n* = 474,467	*n* = 105,717	*n* = 21,379	*n* = 52,665	*n* = 38,093
Age					
65–69	191,377 (40.33)	41,715 (39.46)	8397 (39.28)	21,463 (40.75)	14,228 (37.35)
70–74	135,157 (28.49)	29,717 (28.11)	5873 (27.47)	15,056 (28.59)	10,555 (27.71)
75–79	73,573 (15.51)	16,767 (15.86)	3451 (16.14)	8269 (15.70)	6093 (16.00)
80–84	44,688 (9.42)	10,247 (9.69)	2037 (9.53)	4837 (9.18)	4050 (10.63)
≥85	29,672 (6.25)	7271 (6.88)	1621 (7.58)	3040 (5.77)	3167 (8.31)
Gender					
Male	195,538 (41.21)	47,019 (44.48)	8957 (41.90)	19,830 (37.65)	21,810 (57.25)
Female	278,929 (58.79)	58,698 (55.52)	12,422 (58.10)	32,835 (62.35)	16,283 (42.75)
Census register					
Guangzhou	429,837 (90.59)	96,817 (91.58)	19,721 (92.24)	47,940 (91.03)	35,085 (92.10)
Non-Guangzhou	44,630 (9.41)	8900 (8.42)	1658 (7.76)	4725 (8.97)	3008 (7.90)
Living area					
Urban	318,793 (67.19)	67,268 (63.63)	13,270 (62.07)	34,615 (65.73)	23,159 (60.80)
Rural	155,674 (32.81)	38,449 (36.37)	8109 (37.93)	18,050 (34.27)	14,934 (39.20)
Education level					
No school	48,904 (10.31)	13,054 (12.35)	2758 (12.90)	5779 (10.97)	5475 (14.37)
Primary	152,865 (32.22)	36,389 (34.42)	7275 (34.03)	18,099 (34.37)	13,313 (34.95)
Secondary	175,530 (37.00)	33,752 (31.93)	7280 (34.05)	17,594 (33.41)	10,514 (27.60)
College	97,168 (20.48)	22,522 (21.30)	4066 (19.02)	11,193 (21.25)	8791 (23.08)
Marital status					
Single	54,679 (11.52)	15,044 (14.23)	3102 (14.51)	6247 (11.86)	6964 (18.28)
Married	419,788 (88.48)	90,673 (85.77)	18,277 (85.49)	46,418 (88.14)	31,129 (81.72)
Medical insurance					
Uninsured	19,103 (4.03)	3503 (3.31)	632 (2.96)	1932 (3.67)	1115 (2.93)
Insured	455,364 (95.97)	102,214 (96.69)	20,747 (97.04)	50,733 (96.33)	36,978 (97.07)
Current smoking ^a^					
No	417,037 (87.91)	95,135 (90.00)	17,664 (82.63)	48,427 (91.97)	34,920 (91.67)
Yes	57,377 (12.09)	10,571 (10.00)	3712 (17.37)	4231 (8.03)	3172 (8.33)
Alcohol consumption ^a^					
No	424,755 (89.55)	96,521 (91.33)	19,414 (90.83)	47,427 (90.08)	35,783 (93.96)
Yes	49,563 (10.45)	9167 (8.67)	1961 (9.17)	5223 (9.92)	2299 (6.04)
Dietary habits ^a^					
Balanced	459,908 (97.49)	102,431 (97.47)	20,805 (97.75)	50,810 (97.14)	37,074 (97.86)
Meat or fish diet	2832 (0.60)	689 (0.66)	89 (0.42)	466 (0.89)	162 (0.43)
Vegetarian diet	9007 (1.91)	1970 (1.87)	390 (1.83)	1028 (1.97)	648 (1.71)
Physical activity ^a^					
No	237,550 (51.94)	57,483 (56.39)	12,075 (58.66)	26,160 (51.71)	23,298 (62.98)
Yes	219,831 (48.06)	44,463 (43.61)	8511 (41.34)	24,430 (48.29)	13,694 (37.02)
Hypertension					
No	152,821 (32.21)	32,666 (30.90)	8339 (39.01)	14,151 (26.87)	12,231 (32.11)
Yes	321,646 (67.79)	73,051 (69.10)	13,040 (60.99)	38,514 (73.13)	25,862 (67.89)
Diabetes					
No	360,094 (75.89)	78,840 (74.58)	17,177 (80.35)	38,158 (72.45)	28,282 (74.24)
Yes	114,373 (24.11)	26,877 (25.42)	4202 (19.65)	14,507 (27.55)	9811 (25.76)
Dyslipidemia					
No	180,874 (38.12)	42,336 (40.05)	9613 (44.96)	18,618 (35.35)	17,088 (44.86)
Yes	293,593 (61.88)	63,381 (59.95)	11,766 (55.04)	34,047 (64.65)	21,005 (55.14)

^a^ The sum did not equal the total number because of the existence of missing values.

**Table 2 nutrients-16-01338-t002:** Prevalence of malnutrition and its three patterns.

Characteristics	Malnutrition (%)	Underweight (%)	Obesity (%)	Low Hemoglobin (%)
Overall	22.28	4.51	11.10	8.03
	(22.16–22.40)	(4.45–4.57)	(11.01–11.19)	(7.95–8.11)
Age				
65–69	21.80	4.39	11.22	7.43
	(21.62–21.98)	(4.30–4.48)	(11.08–11.36)	(7.31–7.55)
70–74	21.99	4.35	11.14	7.81
	(21.77–22.21)	(4.24–4.46)	(10.97–11.31)	(7.67–7.95)
75–79	22.79	4.69	11.24	8.28
	(22.49–23.09)	(4.54–4.84)	(11.01–11.47)	(8.08–8.48)
80–84	22.93	4.56	10.82	9.06
	(22.54–23.32)	(4.37–4.75)	(10.53–11.11)	(8.79–9.33)
≥85	24.50	5.46	10.25	10.67
	(24.01–24.99)	(5.20–5.72)	(9.90–10.60)	(10.32–11.02)
Gender				
Male	24.05	4.58	10.14	11.15
	(23.86–24.24)	(4.49–4.67)	(10.01–10.27)	(11.01–11.29)
Female	21.04	4.45	11.77	5.84
	(20.89–21.19)	(4.37–4.53)	(11.65–11.89)	(5.75–5.93)

**Table 3 nutrients-16-01338-t003:** The association rules for three patterns of malnutrition detected by the association rule mining of Apriori algorithm among participants.

Left-Hand Side	Right-Hand Side	Support (%)	Confidence (%)	Lift	Count
{Dietary habits = balanced, Physical activity = no}	{BMI = underweight}	2.59	5.10	1.13	11,769
{Physical activity = no}	{BMI = underweight}	2.64	5.09	1.13	12,012
{Hypertension = yes, Diabetes = yes}	{BMI = obesity}	2.51	13.34	1.21	11,437
{Diabetes = yes}	{BMI = obesity}	3.06	12.67	1.15	13,910
{Dietary habits = balanced, Diabetes = yes}	{BMI = obesity}	2.97	12.63	1.14	13,517
{Hypertension = yes, Dyslipidemia = yes}	{BMI = obesity}	5.13	12.38	1.12	23,307
{Physical activity = no, Hypertension = yes}	{BMI = obesity}	4.20	11.95	1.08	19,080
{Hypertension = yes}	{BMI = obesity}	8.08	11.92	1.08	36,759
{Dietary habits = balanced, Hypertension = yes}	{BMI = obesity}	7.85	11.89	1.08	35,721
{Physical activity = no, Dyslipidemia = yes}	{BMI = obesity}	3.32	11.75	1.06	15,086
{Dyslipidemia = yes}	{BMI = obesity}	7.01	11.56	1.05	31,876
{Dietary habits = balanced, Dyslipidemia = yes}	{BMI = obesity}	6.82	11.52	1.04	31,029
{Physical activity = yes}	{BMI = obesity}	5.34	11.11	1.01	24,291
{Dietary habits = balanced, Physical activity = yes}	{BMI = obesity}	5.19	11.09	1.00	23,592
{Physical activity = no}	{Low hemoglobin = yes}	5.09	9.81	1.21	23,166
{Diabetes = yes}	{Low hemoglobin = yes}	2.09	8.65	1.07	9499
{Dietary habits = balanced}	{Low hemoglobin = yes}	7.92	8.12	1.00	36,000

**Table 4 nutrients-16-01338-t004:** Association factors for underweight and obesity explored by multivariate logistic regression (interaction effect model) ^a^.

Outcome	Interaction	RERI (95% CI)	AP (95% CI)	S (95% CI)	OR (95% CI)
Underweight	Meat or fish diet × Hypertension	−0.24 (−0.65, 0.11)	−0.36 (−1.18, 0.16)	3.54 (−19.59, 18.78)	0.75 (0.49, 1.16)
	Vegetarian diet × Hypertension	0.08 (−0.12, 0.28)	0.10 (−0.15, 0.32)	0.67 (0.22, 2.21)	1.10 (0.88, 1.37)
	Meat or fish diet × Diabetes	−0.12 (−0.51, 0.29)	−0.17 (−1.34, 0.30)	1.57 (−4.67, 11.77)	0.85 (0.48, 1.48)
	Vegetarian diet × Diabetes	−0.02 (−0.26, 0.20)	−0.03 (−0.41, 0.21)	1.16 (−0.23, 5.09)	0.97 (0.75, 1.25)
	Meat or fish diet × Physical activity	−0.25 (−0.64, 0.13)	−0.41 (−1.40, 0.17)	2.82 (−33.86, 20)	0.74 (0.46, 1.17)
	Vegetarian diet × Physical activity	0.09 (−0.11, 0.26)	0.10 (−0.14, 0.27)	0.56 (−0.14, 2.42)	1.10 (0.89, 1.36)
	Physical activity × Hypertension	0.00 (−0.05, 0.05)	0.00 (−0.08, 0.08)	1.00 (0.87, 1.18)	0.96 (0.91, 1.02)
	Hypertension × Dyslipidemia	0.18 (0.14, 0.23)	0.30 (0.22, 0.37)	0.67 (0.63, 0.73)	1.20 (1.13, 1.27)
	Physical activity × Dyslipidemia	0.04 (−0.01, 0.09)	0.06 (−0.01, 0.13)	0.88 (0.78, 1.03)	1.02 (0.96, 1.08)
	Hypertension × Diabetes	0.11 (0.05, 0.17)	0.15 (0.07, 0.24)	0.73 (0.62, 0.86)	1.11 (1.02, 1.20)
	Diabetes × Dyslipidemia	0.13 (0.07, 0.18)	0.18 (0.10, 0.26)	0.72 (0.63, 0.83)	1.12 (1.05, 1.21)
Obesity	Meat or fish diet × Hypertension	0.79 (0.28, 1.29)	0.27 (0.11, 0.41)	1.72 (1.21, 2.61)	1.23 (0.95, 1.58)
	Vegetarian diet × Hypertension	−0.25 (−0.45, −0.05)	−0.18 (−0.34, −0.04)	0.61 (0.40, 0.90)	0.81 (0.70, 0.94)
	Meat or fish diet × Diabetes	0.81 (0.19, 1.48)	0.28 (0.08, 0.43)	1.74 (1.16, 2.53)	1.25 (0.99, 1.59)
	Vegetarian diet × Diabetes	−0.19 (−0.37, 0.00)	−0.16 (−0.35, 0.00)	0.48 (0.07, 0.99)	0.84 (0.72, 0.98)
	Meat or fish diet × Physical activity	−0.84 (−1.34, −0.43)	−0.53 (−0.94, −0.24)	0.42 (0.22, 0.65)	0.63 (0.50, 0.79)
	Vegetarian diet × Physical activity	−0.06 (−0.20, 0.09)	−0.06 (−0.21, 0.08)	0.46 (−0.82, 4.00)	0.94 (0.82, 1.08)
	Physical activity × Hypertension	−0.04 (−0.09, 0.01)	−0.03 (−0.06, 0.01)	0.93 (0.86, 1.01)	0.94 (0.90, 0.99)
	Hypertension × Dyslipidemia	0.00 (−0.05, 0.06)	0.00 (−0.03, 0.03)	1.00 (0.94, 1.09)	0.94 (0.90, 0.98)
	Physical activity × Dyslipidemia	−0.09 (−0.13, −0.04)	−0.07 (−0.11, −0.03)	0.73 (0.64, 0.86)	0.92 (0.88, 0.96)
	Hypertension × Diabetes	0.13 (0.07, 0.20)	0.08 (0.04, 0.12)	1.23 (1.10, 1.38)	1.03 (0.98, 1.09)
	Diabetes × Dyslipidemia	−0.01 (−0.07, 0.05)	−0.01 (−0.05, 0.03)	0.97 (0.87, 1.10)	0.95 (0.91, 0.99)

AP-attributable proportion due to interaction, S-synergy index, OR-odd ratio, CI-confidence interval. ^a^ The model was adjusted for all socio-demographic characteristics. ×, the interaction of two factors.

## Data Availability

The data that supports the findings of this study are available from the corresponding author upon reasonable request due to ethical and privacy concerns.

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
