# Peer review of "The Association of Malnutrition and Health-Related Factors among 474,467 Older Community-Dwellers: A Population-Based Data Mining Study in Guangzhou, China"

_nutrients, 2024, doi:10.3390/nu16091338_

Round 1
Reviewer 1 Report
Comments and Suggestions for Authors
Congratulations to the authors for preparing this interesting study that aimed to exam the prevalence and associated factors of malnutrition in older community-dwellers, and explore the interaction between associated factors.
Enormous study population was an advantage of this paper (474,467 older community-dwellers).
Some minor corrections should be done:
Dietary habits in the paper were divided into three categories: balanced, meat or fish diet, and vegetarian diet. Dietary habits should be described more precisely in the methodology; it is not clear what the „balanced” or „fish hor meat” diet means.
The third logistic regression model was adjusted for all socio-demographic characteristics considered in this study – usually only the variables significant in bivariate analysis are included in the model. The socio-demographic characteristics used in the model should be described.
Table 2. Prevalence of malnutrition and its three patterns. – the units should be added (%).
In discussion the authors wrote: „Another finding of the present study was that diabetes, hypertension and dyslipidemia confirmed to be risk factors of obesity. Along with obesity, both diabetes and cardiovascular disease were indicators of overnutrition”. Since the study is the cross-sectional one it enables verifying only the relationships between nutritional status and diseases. Therefore, it should be also considered that diabetes, hypertension and dyslipidemia may be the consequence of obesity.
The abbreviation below the tables should be added: RERI (95% CI) AP (95% CI) S (95% CI) OR (95% CI).
There are some details that need to be improved in the manuscript as: font size – (discussion, paragraph 2, line 10 and 11), text alignment – (table 1), style (Discussion, last paragraph: „Vegetarian diet was a independent protective factor of low hemoglobin…”).
Reviewer 2 Report
Comments and Suggestions for Authors
1. Treating malnutrition as a combination of underweight, overweight/obese and low haemoglobin is misguided. The pathways leading to underweight-overweight and low haemoglobin will vary and some individuals will be both underweight/overweight and have low haemoglobin.
2. The rationale for including diabetes, hypertension and dyslipidemia is also unclear.
3. A more straightforward paper would undertake a stepwise binary logistic regression analysis for low and normal haemoglobin and test which of the demographic, socioeconomic etc. variables best explain the variation in low and normal haemoglobin. Dietary habits should be removed from the analyses as nearly all report a balanced diet. The amount of variation explained should be added as given the large sample size a small effect may be significant. Interactions can be included in the model.
4. The analyses should then be repeated for underweight, normal and overweight again using a stepwise multinomial regression procedure.
5. One could then undertake separate analyses to see how well the BMI, haemoglobin, demographic, socio-economic etc variables best predict hypertension, diabetes and dyslipidemia separately and/or generate a multimorbidity scale of 0-3 to see which variables best predict single morbidities or multimorbidity
Comments on the Quality of English LanguageAcceptable
